# Plasma Treatment for Cellulose in Tobacco Paper-Base: The Improvement of Surface Hydrophilicity and Mechanical Property

**DOI:** 10.3390/ma15020418

**Published:** 2022-01-06

**Authors:** Zhao Zhang, Quan Shu, Shaolin Ge, Shouhu Xuan

**Affiliations:** 1Anhui Key Laboratory of Tobacco Chemistry, Hefei 230088, China; zhaoz@mail.ustc.edu.cn; 2CAS Key Laboratory of Mechanical Behavior and Design of Materials, Department of Modern Mechanics, University of Science and Technology of China (USTC), Hefei 230027, China; shuq@mail.ustc.edu.cn; 3Research Center of Tobacco and Health, USTC Anhui Tobacco Joint Lab Tobacco Chemistry, CAS University of Science and Technology of China (USTC), Hefei 230052, China

**Keywords:** tobacco paper-base, plasma, hydrophilicity, cellulose, infiltration rate

## Abstract

This paper reports a plasma treatment (PT) method for improving the surface hydrophilicity and mechanical properties of cellulose in reconstituted tobacco paper-base. The absorption and infiltration rates of water droplets on PT-reconstituted tobacco paper-base-15 s were significantly accelerated. Notably, the increased content of methylene and alkyl groups enabled the tobacco paper-base to absorb more useful substrates in the tobacco extract after plasma treatment. In addition, the tensile mechanical performance of reconstituted tobacco was significantly improved after plasma treatment, which indicated that the content of organic matter absorbed by the tobacco paper-base sheet was increased. Moreover, tobacco extract infiltrated on the surface of PT-reconstituted tobacco paper-base reached 37.7° within 30 s, while it reached 79.9° on the reconstituted tobacco paper-base. Finally, the mechanism by which the surface hydrophilicity and mechanical properties of the cellulose in the tobacco paper-base were improved is discussed.

## 1. Introduction

With developments in paper treatment methodology, plasma techniques have attracted increasing attention for surface modification [1,2,3,4,5,6,7]. Depending on the degree of high and low gas molecular ionization and the particle energy size, plasma is divided into high and low temperature forms. Low-temperature plasma is usually in the range of 290–330 K, so can be used for modification of materials such as polymers [8]. The reported research shows that a low temperature plasma mutation breeding technology can be used for training *Chlorella* strains by combining undirected mutagenesis and directional screening [9]. Therefore, low temperature plasma treatment is applicable to agricultural production. Thibodeaux et al., reported that modification of the surface characteristics of polymer fibers by low-temperature plasma could aid in increasing bonding potential [10]. As a result, low-temperature plasma has a wide range of prospects for application and potential value in manufacturing industry, agriculture and industrial production.

Low-temperature plasma includes nitrogen, argon, ammonia, and oxygen plasma treatments. Due to the advantages of little damage, good treatment effects, energy conservation and environmental protection, oxygen plasma treatment has attracted attention for improving cellulose [11,12,13,14,15]. Recently, it was found that oxygen plasma treatment could be applied to the surface modification of cotton fabric to increase surface adsorption and adhesion of the fibers [16]. Esmail et al., reported that the scaffold’s hydrophilicity was significantly increased under oxygen plasma as a surface treatment [17]. Tobacco paper-base is an important part of the tobacco reconstitution industry [18,19,20,21,22]. It can increase the utilization of the tobacco plant and reduce harmful substrates in the cigarette. To improve quality, the chemical components and property of reconstituted tobacco have attracted increasing attention. However, due to the low active surface and weak mechanical property of freshly prepared paper-base, increasing the practical application of reconstituted tobacco still faces challenges, in which improvement of the surface characteristics of cellulose plays a key role. As a non-byproduct physical process, the oxygen plasma method represents a meaningful approach for the treatment of tobacco paper-base sheets to achieve required characteristics. 

Over recent decades, research on the use of reconstituted tobacco in paper-making has received much attention [22,23,24]. In the process of papermaking, the tobacco extract is coated on the surface of the tobacco paper-base, and then the reconstituted tobacco product is formed. However, due to limitations in the conditions of manufacture and high cost, the adsorption of the tobacco extract onto the tobacco paper-base surface is not satisfactory. Therefore, improving the adsorption of the tobacco paper-base surface in a short time has become an urgent demand. Lu et al., showed that decreasing the viscosity of tobacco extract by centrifugation could promote the permeation rate [25]. Furthermore, by increasing the quality of the tobacco paper-base, more pores are formed inside the tobacco paper-base, which is beneficial to its adsorption to tobacco extract [26]. However, the improvement in adsorption, by changing extract viscosity and the quality of the tobacco paper-base, is limited. Therefore, surface treatment of tobacco paper-base by oxygen plasma to improve hydrophilicity and achieve acceleration of infiltration rate, without decreasing the tobacco extract viscosity, to increase tobacco paper-base quality, is urgently required.

A simple and cost-effective plasma treatment (PT) method was applied to improve the surface hydrophilicity and mechanical properties of reconstituted tobacco (PT-reconstituted tobacco). The hydrophilicity of the cellulose in tobacco paper-base was significantly improved under oxygen plasma treatment, and the infiltration rate of water droplets on its surface was clearly accelerated. Scanning electron microscope (SEM) analysis indicated the tobacco sheets still retained a consistent microstructure after oxygen plasma treatment. Furthermore, Fourier transform infrared spectrum and thermogravimetric analysis indicated that the PT-reconstituted tobacco absorbed more substrates in the tobacco extract. Moreover, the tension performance of PT-reconstituted tobacco was also greatly enhanced. 

## 2. Experimental Section

### 2.1. Materials

The reconstituted tobacco paper-base sheets and the tobacco extract (liquid) were supplied by Anhui China Tobacco Reconstituted Tobacco Technology Co., Ltd., Hefei, China. 

### 2.2. Preparation

To meet industrial production conditions, the tobacco extract was diluted by 50% in deionized water and then sonicated for 30 min. After that, the reconstituted tobacco paper-base was immersed in the tobacco extract for 30 s, and then placed in a 70 °C oven for 2 min. The obtained product was defined as reconstituted tobacco (Figure 1(ai)). As shown in Figure 1(aii), the reconstituted tobacco paper-base was treated with oxygen plasma cleaning before immersing. Firstly, the tobacco paper-base was placed in the plasma machine treatment chamber, and then the vacuum pump was turned on to reduce the air pressure of the gas in the treatment chamber below 0.05 Pa. Meanwhile, oxygen was injected into the treatment chamber to ensure that oxygen was the reaction gas in the chamber. Finally, the radio frequency power supply was turned on and the discharge power adjusted after cleaning, and then the reconstituted tobacco paper-base sheets were treated with oxygen plasma. The product obtained after immersing and drying was defined as plasma-treated reconstituted tobacco (PT-reconstituted tobacco). Images of PT-tobacco paper-base, reconstituted tobacco, and PT-reconstituted tobacco are shown in Figure 1b–d. For convenience, oxygen plasma is simply referred to as plasma treatment in this paper.

### 2.3. Characterization

The microstructures of the reconstituted tobaccos were observed by scanning electron microscope (SEM, Philips of Holland, model XL30 ESEM-TMP, Zeiss, Oberkochen, Germany) under 3 kV. The infrared spectra of reconstituted tobacco paper-base, PT-reconstituted tobacco paper-base, reconstituted tobacco, PT-reconstituted tobacco, and tobacco extract were obtained using a Nicolet 8700 Fourier transform infrared (FT-IR) spectrometer (Thermo Scientific Instrument Co., Waltham, MA, USA). The test wavenumber range of tobacco sheet was from 4000 to 500 cm^−1^ with membrane measuring reflectance. Transmittance tests of tobacco extract were conducted with a KBr wafer in the wavenumber range from 4000 to 400 cm^−1^. Thermal stability of the tobacco and reconstituted tobacco was investigated by a DTG-60H and the heating rate was 10 °C/min. The thermogravimetric properties of the samples were tested from room temperature to 800 °C under N_2_ flow for balance and purge gases. The sample weights of reconstituted tobacco paper-base, PT-reconstituted tobacco paper-base, reconstituted tobacco, and PT-reconstituted tobacco were 2.9242 mg, 4.0215 mg, 6.1214 mg, and 5.2366 mg, respectively. The surface composition of the samples was obtained by X-ray photoemission spectroscopy (XPS, ESCALAB250A, Thermo-VG Scientific, Horsham, UK). The tensile test was conducted using a dynamic mechanical analyzer (DMA, ElectroForce 3200, TA Instruments, Eden Prairie, MN 55344, USA). Contact angles and the images of droplets were captured by commercial camera (Nikon, Tokyo, Japan) and high-speed video camera (Nikon, Japan). The rheological property of the tobacco extract was measured by a commercial rheometer (Physica MCR 302, Anton Paar Co., Graz, Austria). The surface of the tobacco paper-base was treated by plasma cleaning machine (WH-1000, Suzhou Wenhao Microfluidic Technology Co., Ltd., Suzhou, China). The ultrasonic treatment was performed using an automatic ultrasonic cleaning machine (SK5200HP, Shanghai Kedao Ultrasonic Instrument Co., Ltd., Shanghai, China) under 120 W working power and 25 °C temperature. The dimensions of the tobacco paper-base sheet for water/tobacco extract droplet infiltration experiments were 37.3 mm × 31.3 mm × 0.2 mm. In the quasistatic tensile test, the dimensions of the tobacco sheet were 20 mm × 4 mm × 5 mm for the central tensile area (ISO 37: 2005).

## 3. Results and Discussion

Plasma treatment changed the surface characteristics of cellulose in the paper-base sheets, thereby improving the infiltration properties of the tobacco sheets. SEM images provided detailed microstructure information of the traditional reconstituted tobacco paper-base and PT-reconstituted tobacco paper-base, showing the morphologies of the cellulose in tobacco paper-base before and after plasma treatment (Figure 2). Figure 2a,b present the SEM images of reconstituted tobacco paper-base sheet and the magnified area of the paper-base sheet, respectively. Figure 2b shows the clear vein structure on the surface of the tobacco paper-base sheet. After plasma treatment, the surface structure of the tobacco sheet remained intact and the inherent mechanical properties of the sheets were also maintained (Figure 2c). Figure 2d represents the enlarged area of the PT-tobacco sheet, which showed no obvious structural difference compared with the reconstituted tobacco paper-base sheet. This was attributed to the weak intensity of the plasma (Figure 2b). Plasma treatment influenced the surface hydrophilicity of the tobacco paper-base sheet, thereby improving the compatibility of the paper-base surface.

XPS spectra were produced to analyze the chemical elements in the reconstituted tobacco paper-base and PT-reconstituted tobacco paper-base (Figure 3). As shown in Figure 3a,b, strong peaks of O 1s and C 1s were clearly observed in the spectra of the reconstituted tobacco paper-base and PT-reconstituted tobacco paper-base. A tiny N 1s peak was observed in the spectra, which may have been due to the presence of protein or total nitrogen. Additionally, Figure 3a,b show the difference between the N 1s peak in reconstituted tobacco paper-base and PT-reconstituted tobacco paper-base. The change in the N 1s peak shows that after plasma treatment more functional groups were doped, which indicates that the tobacco paper-base absorbed more organic matter. According to the XPS results, the K 2p peak was not found in the spectra, and the atomic content was 0% (Figure 3a,b). As shown in Figure 3c, the FT-IR spectrum waveforms of the reconstituted tobacco paper-base and PT-reconstituted tobacco paper-base remained consistent. The peaks of PT-reconstituted tobacco paper-base at 3300 cm^−1^ and 1600 cm^−1^ indicate the presence of O−H bonds. The peak of PT-reconstituted tobacco paper-base at 2900 cm^−1^ shows the stretching vibration of C−H bonds saturated with methylene (Figure 3d). The peak of PT-reconstituted tobacco paper-base at 1400 cm^−1^ shows the bending vibration of C−H bonds saturated with alkyl.

To further investigate the effect of plasma treatment time on the surface hydrophilicity of tobacco paper-base sheets, the infiltration of water droplets on the tobacco paper-base sheet surface at different treatment times was observed (Figure 4). Additionally, the immersion change of water droplets was recorded using a commercial camera. Here, the reconstituted tobacco paper-bases after different plasma treatment times are abbreviated as: PT-tobacco 5 s (Appendix A), PT-tobacco 10 s (Appendix A), PT-tobacco 15 s (Appendix A) and PT-tobacco 20 s (Appendix A). The hydrophilicity of the cellulose surface was characterized and analyzed using contact angle technology. Figure 4a shows the infiltration process of a water droplet on the surface of reconstituted tobacco paper-base without plasma treatment (Appendix A). After 10 s, the reconstituted tobacco paper-base sheets still did not completely absorb the droplet, and the water droplet remained on the surface. Figure 5a exhibits the contact angle change of a water droplet on the reconstituted tobacco paper-base surface vs. time; the contact angle decreased from 83.2° to 13.9° within 10 s. Figure 4b shows the absorption of the water droplet on the PT-reconstituted tobacco paper-base-5 s surface. The contact angle of the water droplet on the PT-reconstituted tobacco paper-base-5 s surface decreased from 75.7° to 7.8° (Figure 5a). Compared with the water infiltration of the reconstituted tobacco paper-base, the hydrophilicity of the PT-reconstituted tobacco paper-base-5 s was significantly improved.

As shown in Figure 4c, the water droplet was rapidly absorbed on the surface of the PT-reconstituted tobacco paper-base-10 s, and the contact angle of the water droplet on its surface decreased from 74.4° to 20.9° within 0.3 s (Figure 5b). The water droplet infiltration on the tobacco surface became quicker with increasing plasma treatment time. Figure 4d shows the process of water droplet absorption on the PT-reconstituted tobacco paper-base-15 s; the infiltration rate of the water droplet was significantly accelerated. The contact angle of the water droplet on the PT-reconstituted tobacco paper-base-15 s decreased from 72.4° to 14.3° within 0.3 s (Figure 5b). Here, the mean infiltration rate (v) is defined as
(1)$$v=\left(\theta_{1}\right.-\left.\theta_{2}\right)/t$$
where $\theta_{1}$ and $\theta_{2}$ is the initial contact angle (such as 72.4°) and final contact angle (such as 14.3°), respectively. Additionally, t is the time of the contact angle infiltration process (such as 0.3 s).

According to the experimental results, the mean infiltration rates of reconstituted tobacco paper-base and PT-reconstituted tobacco paper-base-15 s were 6.9°/s and 193.5°/s, respectively (Figure 5d). Clearly, the hydrophilicity of the tobacco was greatly improved, and the mean infiltration rate of water droplets on the PT-reconstituted tobacco paper-base-15 s surface was 28 times that of the reconstituted tobacco paper-base. Figure 4e shows the absorption process of a water droplet on the PT-reconstituted tobacco paper-base-20 s surface; the contact angle of the water droplet decreased from 68.6° to 6.8° within 3 s (Figure 5b). Additionally, the mean infiltration rate of the PT-reconstituted tobacco paper-base-20 s was 205.8 °/s (Figure 5d). Here, it was found that the improvement in the cellulose surface hydrophilicity was not significant when the plasma treatment time exceeded 15 s. Moreover, the final plasma treatment time was 15 s when considering the time cost of real-world factory application. Figure 5c presents a comparison of the contact angle changes of a water droplet on the reconstituted tobacco paper-base and PT-reconstituted tobacco paper-base-15 s surface, respectively. Plasma treatment for 15 s greatly improved the hydrophilicity of the tobacco surface and significantly shortened the infiltration process of the water droplet.

Based on the above analysis, the optimal time for plasma treatment was determined to be 15 s. Moreover, the application of plasma technology to the production of reconstituted tobacco was thoroughly explored. The detailed microstructure of the reconstituted tobacco and PT-reconstituted tobacco are shown in Figure 6. The reconstituted tobacco paper-base sheet was first immersed in the tobacco extract, and then the reconstituted tobacco was formed after drying (Figure 1(ai)). Figure 6a shows the microstructure of the reconstituted tobacco. The magnified image of the reconstituted tobacco presents the microstructure of the combined concentrate and sheets (Figure 6b). The reconstituted tobacco paper-base sheet was treated by plasma and then immersed into the tobacco extract to obtain the PT-reconstituted tobacco (Figure 1(aii)). Figure 6c shows the microstructure of the PT-reconstituted tobacco. Plasma cleaning modified the tobacco paper-base surface, and the microstructure of the reconstituted tobacco remained consistent (Figure 6a,c). Figure 6d presents the enlarged area of the tobacco extract, which shows that the tobacco extract was tightly attached to the surface of the sheet after drying. The dried organic tobacco extract ensured the aroma and taste of the reconstituted tobacco.

The chemical components in tobacco sheets are the basic factors that affect the quality of tobacco sheets and the main influencing factors determining the quality of smoke. The raw materials of reconstituted tobacco are generally divided into three categories, including tobacco stems, fragments and tobacco dust. Normally, the chemical evaluation indexes of tobacco sheets include reducing sugar, starch and protein, etc. Here, the XPS spectra and FT-IR spectrum of reconstituted tobacco sheet are shown in Figure 7. After immersing the tobacco paper-base into the tobacco extract, the O 1 s and C 1 s peaks could be clearly observed; the intensity of the C 1s peaks was enhanced in the spectra of the reconstituted tobacco and PT-reconstituted tobacco (Figure 7a,b). The C atomic content of the reconstituted tobacco paper-base was 80.3%. Furthermore, the C atomic content of PT-reconstituted tobacco increased by 5.2 and 1.4% compared to the reconstituted tobacco paper-base and reconstituted tobacco, respectively. The results show that the PT-reconstituted tobacco absorbed more extracted substrates compared to the reconstituted tobacco. Clearly, the plasma treatment greatly improved the hydrophilicity of the tobacco sheet surface, so that the coating efficiency of the tobacco extract on its surface was obviously enhanced.

The infrared spectrum waveforms of tobacco extract are shown in Figure 7c. The peaks of tobacco extract at 3300 cm^−1^ and 1600 cm^−1^ indicate the presence of O−H bonds. Figure 7d shows the infrared wavelength of the reconstituted tobacco and PT-reconstituted tobacco. The peaks of PT-reconstituted tobacco at 1000 cm^−1^ and 1120 cm^−1^ correspond to the stretching vibration of C−O bonds (Figure 7d). By comparison, from the peaks of tobacco at 2900 cm^−1^, and of 1380 cm^−1^ and 1000 cm^−1^ of PT-reconstituted tobacco and reconstituted tobacco, respectively, it is clear that the plasma treatment improves the tobacco extract content absorbed by the tobacco sheets during the soaking process. Moradi Y et al. reported the plasma modification of beta-carotene-loaded nanofibers to enhance osteogenic differentiation, and the FTIR and contact angle measurements were used to detect and confirm surface chemical changes. Treatment by plasma made the 1088 cm^−1^ peak in the FTIR more intense. The bands located in the regions of 2943 cm^−1^ and 2869 cm^−1^ represent asymmetric CH_2_ stretching, and symmetric CH_2_ stretching, respectively, which are observed to be more intense after plasma modification [27]. Here, the O−H bond content at the peaks of 3300 and 1600 cm^−1^ of PT-reconstituted tobacco was higher than that of reconstituted tobacco. Hence, plasma treatment greatly accelerated the infiltration rate in the same immersion time.

The quality stability of reconstituted tobacco products includes chemical quality stability and physical quality stability. Additionally, the burning rate is an important index to characterize the physical quality of tobacco sheet. Here, the thermal stability of various tobacco sheets was investigated by Thermogravimetric analysis (TGA) (Figure 8). The reconstituted tobacco paper-base and PT-reconstituted tobacco paper-base showed highly similar TGA curves (Figure 8a,b). Moreover, the weight loss could be divided into two stages. The first stage occurred before 100 °C; the weight loss at this stage was attributed to the evaporation of water in the tobacco. The temperature of the second stage changed from 165 °C to 392 °C; the weight loss could be ascribed to the pyrogenic decomposition of organic matter (such as sugar and protein) in the tobacco. Here, plasma was treated on the tobacco surface and had no effect on the thermal stability of the tobacco (Figure 8a,b).

Figure 8c,d show the TGA curve of the reconstituted tobacco and PT-reconstituted tobacco, in which the weight loss could be divided into three stages. The first stage occurred before 100 °C; the weight change was mainly due to the evaporation of water within the tobacco (Figure 8c,d). The second stage occurred between 129 °C and 281 °C; the weight loss was mainly because of the thermal decomposition of the tobacco extract adsorbed on the reconstituted tobacco paper-base surface (Figure 8c). The third stage occurred at 356 °C, which was due to the decomposition of the organic matter in the reconstituted tobacco, indicating good thermal stability of the reconstituted tobacco (Figure 8c,d). In contrast to the reconstituted tobacco, the second stage of PT-reconstituted tobacco changed from 129 °C to 259 °C (Figure 8d). According to the temperature change, it is assumed that the improvements derived from the plasma treatment. The surface of the PT-tobacco possessed good hydrophilicity, as it absorbed more organic components during the same soaking time, which led to a decrease in the thermal decomposition temperature.

The mechanical performance of the reconstituted tobacco paper-base and reconstituted tobacco was also explored (Figure 9). The tensile fracture process was conducted using a DMA. Figure 9a presents the tensile fracture process of reconstituted tobacco paper-base and the inset is the optical image of the reconstituted tobacco paper-base. The maximum tensile fracture force of reconstituted tobacco paper-base was tested to be 2.5 N with 0.5 mm tensile displacement. Figure 9b shows the tensile fracture of reconstituted tobacco; the break force was 3.8 N under 0.3 mm tensile displacement. The reconstituted tobacco absorbed the tobacco extract, which led to increase in the content of protein, sugar and other organic matter. Figure 9c shows the tension performance of PT-reconstituted tobacco; the fracture force was 4.4 N at 0.6 mm displacement. Additionally, the maximum tensile fracture force of the reconstituted tobacco paper-base, reconstituted tobacco and PT-reconstituted tobacco is presented in Figure 9d. Here, the breaking force of PT-reconstituted tobacco was the maximum. The plasma treatment accelerated the infiltration rate of the tobacco extract, thus more organic matter was absorbed into the tobacco at the same time, enhancing intermolecular forces.

To further investigate the infiltration behavior of tobacco extract on the tobacco paper-base surface, the rheological property of the tobacco extract was tested (Figure 10). The tobacco extract was diluted (50 wt. %) before characterizing the performance to satisfy the concentration standard of the tobacco extract in actual production. Figure 10a shows the images of the tobacco extract placed on the test platform at temperatures of 25 °C and 70 °C, respectively. It was found that the tobacco extract became molten due to the high temperature at 70 °C. Figure 10b shows the viscosity of the tobacco extract when shear rates changed from 0.01 s^−1^ to 1000 s^−1^. The viscosity of the extract decreased with increasing in shear rate. The viscosities varied from 627.5 Pa·s to 4.8 Pa·s and from 388,793.1 Pa·s to 5.6 Pa·s under 25 °C and 70 °C, respectively. Figure 10c shows the relationship between the mean viscosity of the tobacco extract and temperature in the shear rate range of 100–1000 s^−1^. The viscosity of the tobacco extract reached its maximum at 70 °C. In addition, the viscosity of the tobacco extract at 70 °C and shear rate of 10 s^−1^ was 476.7 Pa·s (Figure 10d). Since the hydrogen bond between the water molecules of the tobacco extract was damaged at 70 °C, the water volatilization was accelerated and then the viscosity of the extract increased. Furthermore, the concentration of tobacco extract varied slightly at a fixed shear rate of 10 s^−1^ when the temperature ranged from 25 °C to 60 °C.

After exploring the plasma treatment time and the tobacco extract rheology performance, the infiltration behavior of tobacco extract on the tobacco paper-base surface was investigated. To clearly record the infiltration of the tobacco extract on the tobacco paper-base surface, the process of tobacco paper-base sheet absorption of droplets was captured by a high-speed camera (Figure 11). The infiltration of the tobacco extract on the surface of the reconstituted tobacco paper-base and PT-reconstituted tobacco paper-base is shown in Figure 11a (Appendix A) and Figure 11b (Appendix A), respectively. According to the high-speed camera images, the infiltration rate of the tobacco extract on the surface of PT-reconstituted tobacco paper-base was significantly faster than that on the reconstituted tobacco paper-base surface (Figure 11a,b). When the absorption time was 15 s, the contact angle of the extract reached 83.2° at the reconstituted tobacco paper-base surface and 50.6° at the PT-reconstituted tobacco paper-base surface. 

The contact angle of the extract infiltration on different tobacco paper-base surfaces is presented in Figure 11c,d. The contact angle of the tobacco extract was 79.9° when infiltrated at the reconstituted tobacco paper-base for 30 s and reached 37.7° for the PT-reconstituted tobacco paper-base. Moreover, the mean infiltration rates of the tobacco extract at 30 s on the reconstituted tobacco paper-base and PT-reconstituted tobacco paper-base surface were 0.2°/s and 1.7°/s, respectively (Figure 11c,d inset images). After 100 s, the contact angle of the tobacco extract on the reconstituted tobacco paper-base was 70.3°, while it reached 30.8° on the PT-reconstituted tobacco paper-base surface (Figure 11e). Obviously, the hydrophilicity of the cellulose in the tobacco paper-base was greatly enhanced after the plasma treatment. Therefore, the infiltration rate of the tobacco extract on the tobacco paper-base surface was accelerated, and the absorption time was shortened. 

## 4. Conclusions

In summary, a plasma treatment process was applied to reconstituted tobacco paper-base to improve the hydrophilicity of the cellulose surface. After 15 s plasma treatment, the hydrophilicity of the tobacco surface was significantly improved, and the infiltration process of water droplets was significantly shortened. Furthermore, the FT-IR spectrum showed that the organic functional groups, such as methylene, alkyl, and hydroxyl, were significantly increased in the PT-reconstituted tobacco. According to the TGA curves, the decomposition temperature range (129–259 °C) of the PT-reconstituted tobacco was lower than that of reconstituted tobacco due to the greater absorption of organic matter. In addition, the O 1s and C 1s peaks in XPS spectra of the reconstituted tobacco and PT-reconstituted tobacco could be clearly observed, and the intensity of the C 1s peaks was enhanced in the PT-reconstituted tobacco spectra. Moreover, the maximum tension fracture force of the PT-reconstituted tobacco was increased by the plasma treatment. Moreover, the contact angle of the tobacco extract on the surface of the PT-tobacco paper-base dropped rapidly within 30 s, and the infiltration rate was greatly improved. In sum, the plasma treatment method shows significant potential for improving the quality of tobacco paper-base, and elsewhere in the paper-base industry.

## Figures and Tables

**Figure 1 materials-15-00418-f001:**
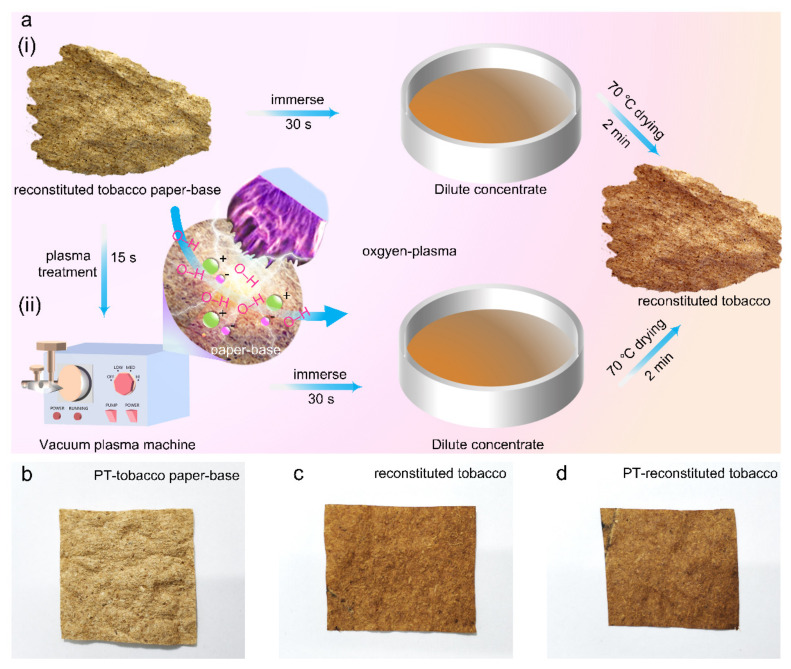
Preparation process of (**a**) (**i**) reconstituted tobacco and (**ii**) plasma-treated (PT) reconstituted tobacco; images of (**b**) PT-tobacco paper-base, (**c**) reconstituted tobacco, and (**d**) PT-reconstituted tobacco.

**Figure 2 materials-15-00418-f002:**
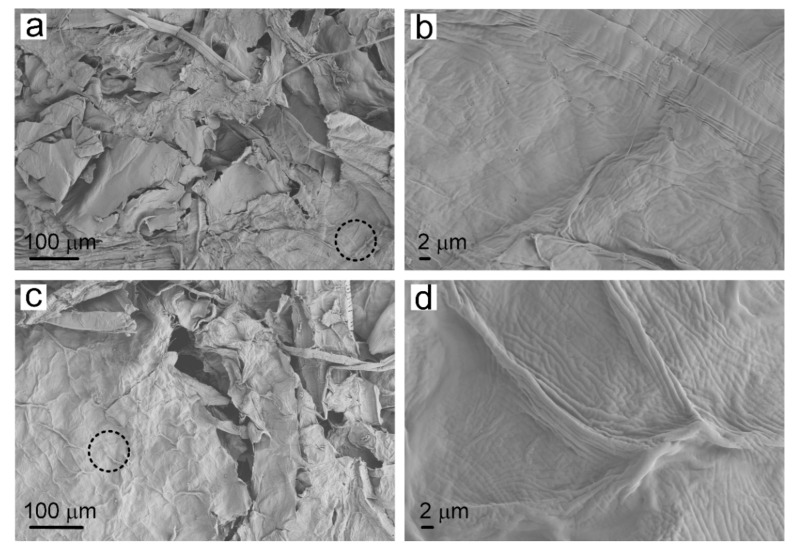
SEM images of (**a**) reconstituted tobacco paper-base and (**b**) the magnified image of the paper-base sheet; SEM images of (**c**) PT-tobacco paper-base and (**d**) the enlarged area of the PT-paper-base sheet.

**Figure 3 materials-15-00418-f003:**
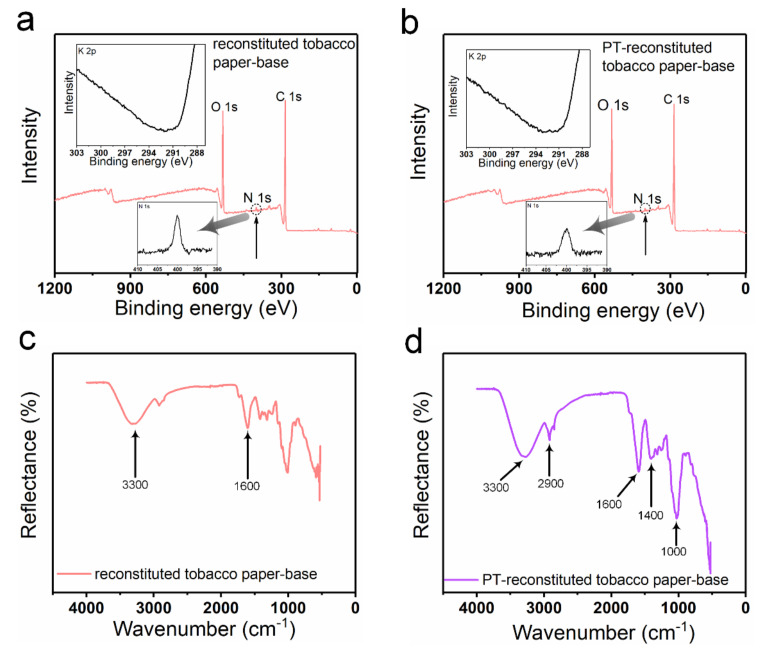
X-ray photoelectron spectroscopy (XPS) of (**a**) reconstituted tobacco paper-base and (**b**) PT-reconstituted tobacco paper-base; Inset: high-resolution spectra for potassium and nitrogen of broad-range XPS spectra. FT-IR spectra of (**c**) reconstituted tobacco paper-base and (**d**) PT-reconstituted tobacco paper-base.

**Figure 4 materials-15-00418-f004:**
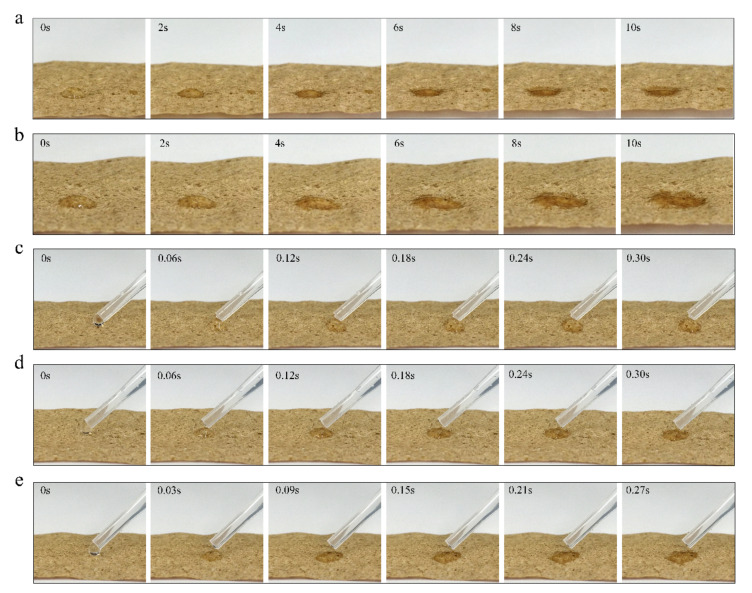
Infiltration of water droplets on the reconstituted tobacco paper-base surface with different plasma treatment time: (**a**) 0 s, (**b**) 5 s, (**c**) 10 s, (**d**) 15 s, and (**e**) 20 s.

**Figure 5 materials-15-00418-f005:**
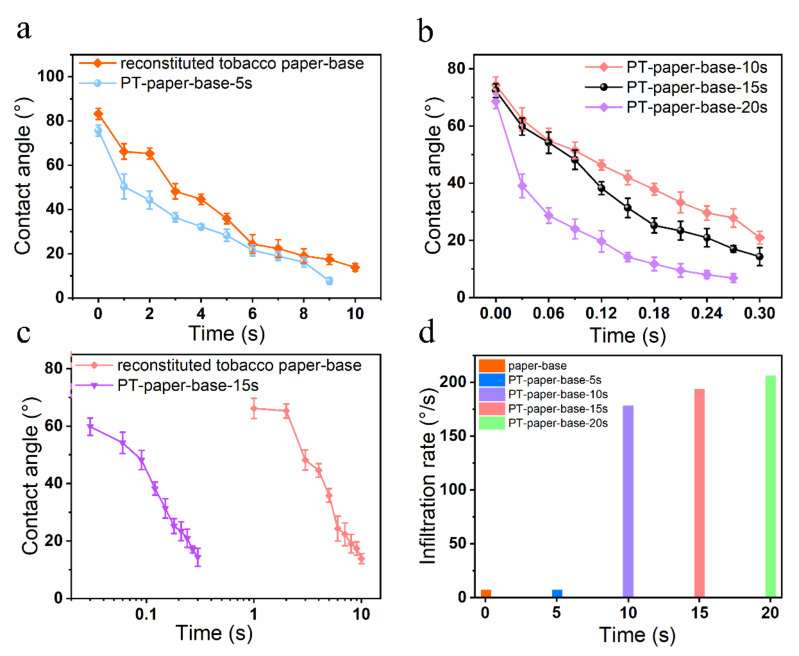
The infiltration contact angle of a water droplet on the surface of different tobacco paper-base sheets. The contact angle comparison of (**a**) reconstituted tobacco paper-base and PT-tobacco-5s, (**b**) PT-tobacco-10 s, PT-tobacco-15 s and PT-tobacco-20 s, (**c**) reconstituted tobacco paper-base and PT-tobacco-15 s; Different mean infiltration rate of (**d**) reconstituted tobacco paper-base, PT-tobacco-5 s, PT-tobacco-10 s, PT-tobacco-15 s and PT-tobacco-20 s.

**Figure 6 materials-15-00418-f006:**
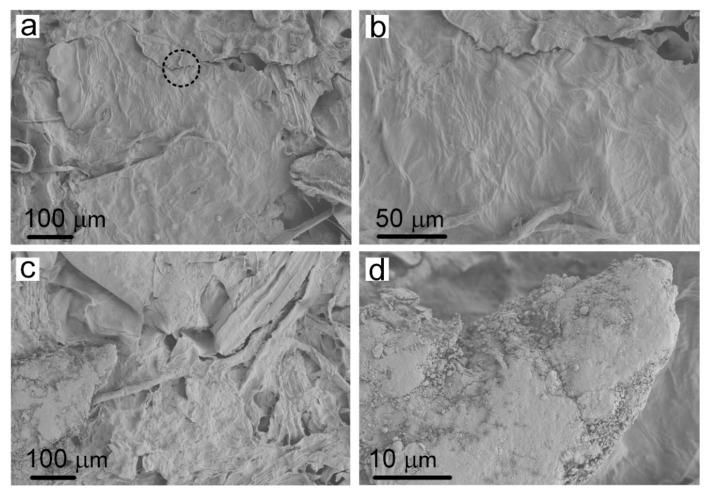
SEM images of: (**a**) the reconstituted tobacco and (**b**) the magnified image of reconstituted tobacco sheet; (**c**) PT-reconstituted tobacco and (**d**) the dried organic tobacco extract adhering to the PT-reconstituted tobacco surface.

**Figure 7 materials-15-00418-f007:**
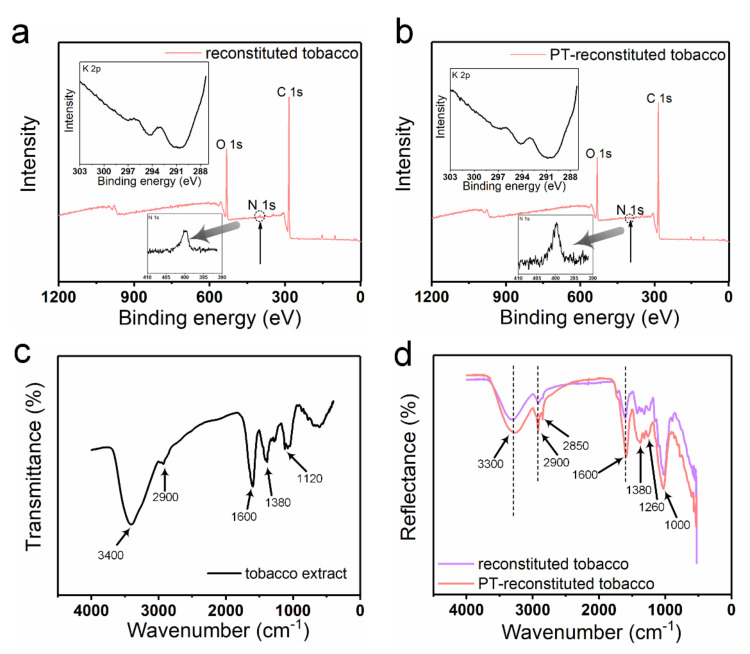
X-ray photoelectron spectroscopy (XPS) of (**a**) reconstituted tobacco and (**b**) PT-reconstituted tobacco; Inset: the high-resolution spectra for potassium and nitrogen of broad-range XPS spectra. FT-IR spectra of (**c**) tobacco extract, and (**d**) the comparison of reconstituted tobacco and PT-reconstituted tobacco.

**Figure 8 materials-15-00418-f008:**
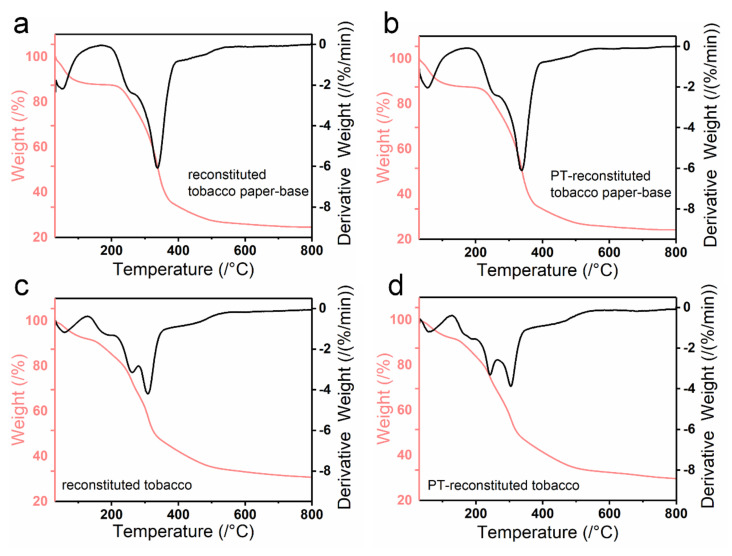
Thermogravimetric analysis (TGA) curves of (**a**) reconstituted tobacco paper-base, (**b**) PT-reconstituted tobacco paper-base, (**c**) reconstituted tobacco, and (**d**) PT-reconstituted tobacco.

**Figure 9 materials-15-00418-f009:**
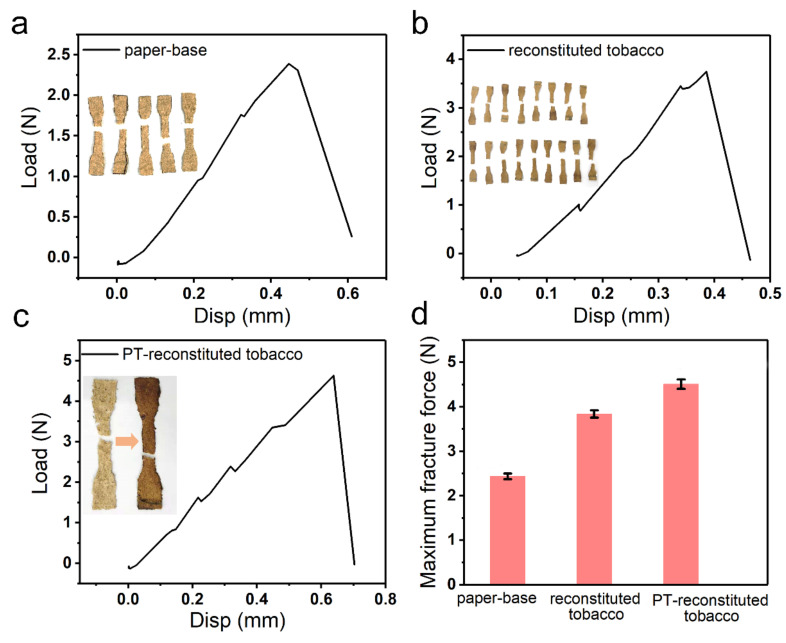
Tensile performance of different tobacco sheets. Tensile fracture behavior of (**a**) reconstituted tobacco paper-base, (**b**) reconstituted tobacco, and (**c**) PT-reconstituted tobacco. The maximum tensile fracture force of (**d**) reconstituted tobacco paper-base, reconstituted tobacco and PT-reconstituted tobacco. Inset images: parallel experiments of the tensile fracture in multiple samples.

**Figure 10 materials-15-00418-f010:**
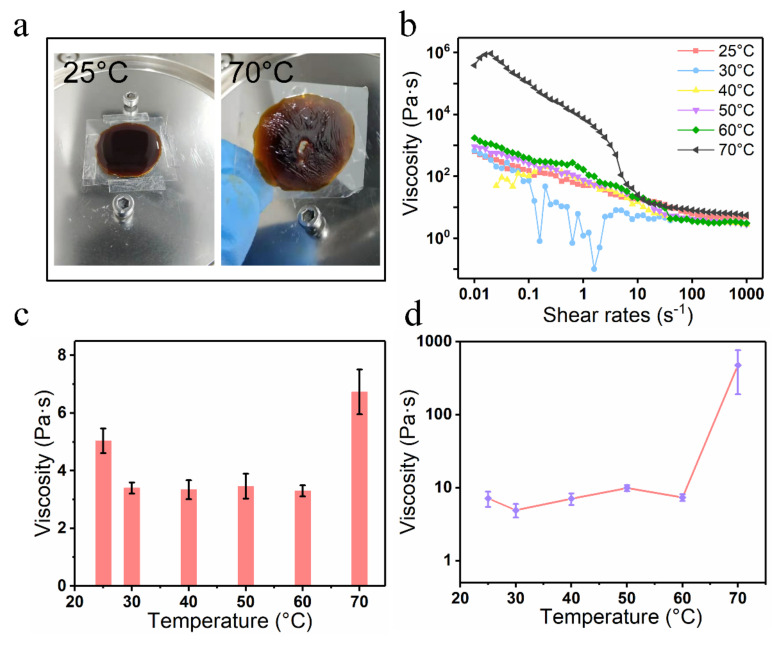
Rheology property of the tobacco extract (liquid) after dilution. The images of tobacco extract (**a**) at 25 °C and 70 °C; (**b**) Viscosity of tobacco extract under different temperature vs. shear rates; (**c**) The mean viscosity changes vs. temperature at shear rate range of 100–1000 s^−1^; (**d**) Viscosity changes vs. temperature at shear rate of 10 s^−1^.

**Figure 11 materials-15-00418-f011:**
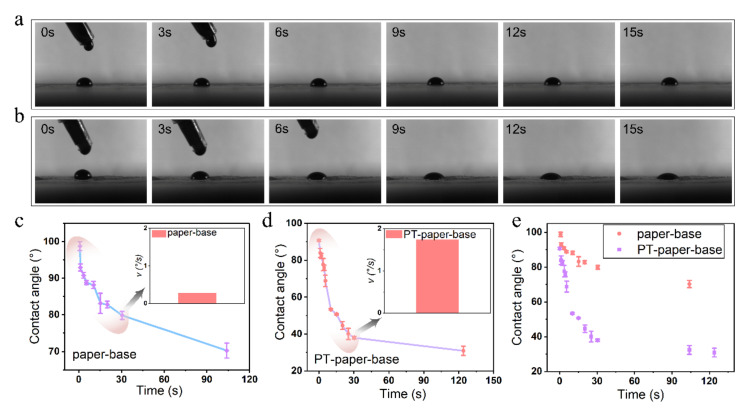
Infiltration of the tobacco extract on different tobacco paper-base surface. High-speed camera images of tobacco extract absorption on the (**a**) tobacco paper-base and (**b**) PT-tobacco paper-base surface; The infiltration contact angle of tobacco extract on the (**c**) tobacco paper-base and (**d**) PT-tobacco paper-base surface vs. time; (**e**) The infiltration contact angle comparison of the tobacco extract on the tobacco paper-base and PT-tobacco paper-base surface. Inset: the mean infiltration rates of the tobacco extract on the (**c**) reconstituted tobacco paper-base and (**d**) PT-reconstituted tobacco paper-base surface.

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
