# Peer review of "Plasma Treatment for Cellulose in Tobacco Paper-Base: The Improvement of Surface Hydrophilicity and Mechanical Property"

_materials, 2022, doi:10.3390/ma15020418_

Round 1

Reviewer 1 Report

This is high quality work that can be published as-is. However, there are still some grammar errors in text and the manuscript should be polished more.

Author Response

Reply to reviewer #1.Question 1: This is high quality work that can be published as-is. However, there are still some grammar errors in text and the manuscript should be polished more. Our reply: Thanks for your review. The relevant contents were revised and highlighted in the revision. (Page 1/15, Abstract and Introduction section; Page 2/15, Introduction and Experimental section; Page 4/15, results discussion of Figure 2; Page 5/15, results discussion of Figure 3; Page 6/15, results discussion of Figure 4; Page 9/15-10/15, results discussion of Figure 7c and 7d; Page 11/15-12/15, results discussion of Figure 9 and Figure 10; Page 13/15, results discussion of Figure 11; Page 14/15, Conclusion section.)

Reviewer 2 Report

Dear authors, I very respectfully present my considerations only correct to better understand the state of the art and, if possible, contribute:

 page 4/15 in Figure 2, images by SEM, I do not consider a plausible argument to show that surface changes actually occurred. Despite the magnification being at 30 micrometers in the two magnifications of the chosen regions, the working distance between the electron beam and the sample was emitted must be taken into account. It needs to be fixed in all comparative measures. For these arguments proposed, larger extensions could bring better contributions or proofs. In our experience with cellulose, close to nanometer magnifications revealed interesting plasma effects on cellulosic fiber structures. It is known that when trying to amplify the electron beam, it degrades the cellulosic fibers, which is why AFM images are more revealing. However I believe that it is possible by SEM to make images with higher magnifications than this limit that was presented. Could you argue about that?

 page 5/15 in Figure 3, the FTIR spectra as they are being presented do not allow to notice significant changes, perhaps, magnifications of the regions of interest would be adequate. As well as evaluating the ratio between the intensities of internal bands to a spectrum to eliminate the concentration effect, and thus, it will be possible to better compare the effects when comparing the obtained values. At the end of the article, in the Conclusion item, it was mentioned: "Furthermore, the FT-IR spectrum shows that the organic functional groups such as methylene, alkyl, and hydroxyl are significantly increased in the PT-reconstituted tobacco."

     In the FT-IR spectrum it is difficult to make such an assertion due to the large overlapping of signals in the comprehended region. Could you argue about that?

 page 10/15: in “Obviously, the O−H bonds content at the peak of 1600 cm-1 of PT-reconstituted tobacco is higher than that of reconstituted tobacco”. As already mentioned, the intensity of the signals are proportional to the concentrations and the molar absorptivity. This is an example of the need to make relationships between peak or band intensities within the same spectrum so that it can then be used to compare with other FT-IR spectra, for samples with and without plasma treatment. Could you do the intensities calculations and put them in a Table?

 In discussions involving FTIR, literature references were not placed to validate or confront Plasma treatment. If the work is unpublished in relation to this treatment, please impact this in the text. If there are literatures that performed similar treatments, please cite them and compare them with current work.

 page 6/15 for contact angle (CA) it would be appropriate to have information about the average pore size and more information about the volumes of liquid droplets that have been deposited on the papers to better understand the droplet absorption processes, for both, before and after plasma treatments. It is easily noticeable that the plasma treatment changed the surface of the tobacco fibers, however, it does not allow to deduce by CA whether the effect is only due to hydrophilicity or whether the plasma conditions could have modified aspects involving capillary effect, which could also be contribute to accelerating the absorption speed of the drops. Could you argue about that?

 What is the percentage chemical composition of the components present in the tobacco paper-base? If it is a lignocellulosic material, why is only the chemical modification of cellulose being considered? If other components present are also modified, the resulting hydrophilicity is also due to them. Could you argue about that?

 Is it possible to show a chemical modification mechanism via Plasma or pertinent chemical equations?

 Better describe the conditions under which the TGA analysis was performed, that is, flow, type of gas (N2, Air or ???), weighted sample mass.

 Likewise, better describe how the FT-IR analyzes were carried out.

Author Response

Reply to reviewer #2

Dear authors, I very respectfully present my considerations only correct to better understand the state of the art and, if possible, contribute:

Question 1: Page 4/15 in Figure 2, images by SEM, I do not consider a plausible argument to show that surface changes actually occurred. Despite the magnification being at 30 micrometers in the two magnifications of the chosen regions, the working distance between the electron beam and the sample was emitted must be taken into account. It needs to be fixed in all comparative measures. For these arguments proposed, larger extensions could bring better contributions or proofs. In our experience with cellulose, close to nanometer magnifications revealed interesting plasma effects on cellulosic fiber structures. It is known that when trying to amplify the electron beam, it degrades the cellulosic fibers, which is why AFM images are more revealing. However I believe that it is possible by SEM to make images with higher magnifications than this limit that was presented. Could you argue about that?Our reply: Good comment. Here, SEM images provide more detailed microstructure information of the traditional reconstituted tobacco paper-base and PT-reconstituted tobacco paper-base, showing the morphological of the cellulose in tobacco paper-base before and after plasma treatment (Figure 2). Figure 2d represents the enlarged area of the PT-tobacco sheet, which shows no structural difference compared with the reconstituted tobacco paper-base sheet (Figure 2b). Therefore, it is difficult to observe the difference before and after plasma treatment by micromorphology. The high resolution SEM images were added in the revision to replace the unclear images. It must be right nanometer magnifications revealed the plasma effects on cellulosic fiber structures. The difference is hard to be observed in the SEM images which may be responded for the low density treatment. Inspired by this valuable suggestion, the AFM images will be conducted to reveal the effect. Thanks for your good comment again. (Page 4/15, Figure 2b and 2d) Question 2: Page 5/15 in Figure 3, the FTIR spectra as they are being presented do not allow to notice significant changes, perhaps, magnifications of the regions of interest would be adequate. As well as evaluating the ratio between the intensities of internal bands to a spectrum to eliminate the concentration effect, and thus, it will be possible to better compare the effects when comparing the obtained values. At the end of the article, in the Conclusion item, it was mentioned: "Furthermore, the FT-IR spectrum shows that the organic functional groups such as methylene, alkyl, and hydroxyl are significantly increased in the PT-reconstituted tobacco."In the FT-IR spectrum it is difficult to make such an assertion due to the large overlapping of signals in the comprehended region. Could you argue about that?Our reply: Thanks for your suggestion. In Figure 3, the peaks of PT- reconstituted tobacco paper-base at 3300 cm-1 and 1600 cm-1 indicate the presence of O − H bonds. The peak of PT-reconstituted tobacco paper-base at 2900 cm-1 shows the stretching vibration of C − H bonds saturated with methylene (Figure 3d). The peak of PT-reconstituted tobacco paper-base at 1400 cm-1 shows the bending vibration of C − H bonds saturated with alkyl. Moreover, by comparison the peaks of tobacco at 2900 cm-1, 1380 cm-1 and 1000 cm-1 of PT-reconstituted tobacco and reconstituted tobacco, it is clear that the plasma treatment improves the tobacco extract content absorbed by the tobacco sheets during the soaking process (Figure 7d). Actually, it is very right that the FTIR can clearly provide the detail structure change due to the overlapping of the signals. Here, the above results are obtained by an overall consideration of the changes in the FTIR. (Page 5/15, Figure 3c and 3d; Page 9/15, Figure 7d) Question 3: Page 10/15: in “Obviously, the O−H bonds content at the peak of 1600 cm-1 of PT-reconstituted tobacco is higher than that of reconstituted tobacco”. As already mentioned, the intensity of the signals are proportional to the concentrations and the molar absorptivity. This is an example of the need to make relationships between peak or band intensities within the same spectrum so that it can then be used to compare with other FT-IR spectra, for samples with and without plasma treatment. Could you do the intensities calculations and put them in a Table?Our reply: Thanks for your careful review. The reflectance of reconstituted tobacco and PT- reconstituted tobacco at 1600 cm-1 peak is 78.7% and 60.2%, respectively. Similarly, the reflectance of these two tobacco sheet is 78% and 71.5% at 3300 cm-1 peak, respectively. Clearly, plasma treatment makes the 1600 and 3300 cm-1 peak of PT- reconstituted tobacco more intense. The results show that the plasma treatment improves the absorption rate of tobacco paper-base. Question 4: In discussions involving FTIR, literature references were not placed to validate or confront plasma treatment. If the work is unpublished in relation to this treatment, please impact this in the text. If there are literatures that performed similar treatments, please cite them and compare them with current work.Our reply: Thanks for your careful review. Moradi Y et al. reported the plasma modification on Beta-carotene-loaded nanofibers to enhance osteogenic differentiation, and the FTIR and contact angle measurements were used to detect and confirm surface chemical changes. Treatment by plasma makes the 1088 cm-1 peak in the FTIR more intense. The band located in the regions of 2943 cm-1 and 2869 cm-1 represents asymmetric CH2 stretching and symmetric CH2 stretching respectively, which was observed to be more intense after plasma modification [27]. The relevant literature was cited and added in the revision. (Page 9/15 and 10/15)[27] Moradi, Y.; Atyabi, S. A.; Ghiassadin, A.; Bakhshi, H.; Irani, S.; Atyabi, S. M.; Dadgar, N., Cold atmosphere plasma modification on Beta-carotene-loaded nanofibers to enhance osteogenic differentiation. Fibers Polym. 2021. https://doi.org/10.1007/s12221-021-0033-y. Question 5: Page 6/15 for contact angle (CA) it would be appropriate to have information about the average pore size and more information about the volumes of liquid droplets that have been deposited on the papers to better understand the droplet absorption processes, for both, before and after plasma treatments. It is easily noticeable that the plasma treatment changed the surface of the tobacco fibers, however, it does not allow to deduce by CA whether the effect is only due to hydrophilicity or whether the plasma conditions could have modified aspects involving capillary effect, which could also be contribute to accelerating the absorption speed of the drops. Could you argue about that?Our reply: Good suggestion. In this work, the change of contact angle with time is only used to directly reflect the infiltrate rate of the droplets on the surface of the paper-base. Recently, it was found that the oxygen plasma treatment could be applied for surface modification of the cotton fabric to increase the surface adsorption and adhesion of the fibers [16]. As shown in Figure 4, the infiltration rate of water droplets on tobacco paper-base surface gradually accelerated with increasing plasma treatment time.[16] Kert, M.; Forte Tavčer, P.; Hladnik, A.; Spasić, K.; Puač, N.; Petrović, Z. L.; Gorjanc, M., Application of fragrance microcapsules onto cotton fabric after treatment with oxygen and nitrogen plasma. Coatings 2021, 11, (10), 1181. Question 6: What is the percentage chemical composition of the components present in the tobacco paper-base? If it is a lignocellulosic material, why is only the chemical modification of cellulose being considered? If other components present are also modified, the resulting hydrophilicity is also due to them. Could you argue about that?Our reply: Good comment. The specific chemical composition of components in tobacco paper-base is relatively complex, and it has not been studied systematically yet. We will focus on the relevant research in future work. As we all know, the lignocellulosic belongs to cellulose. Due to the complex composition of tobacco paper-base, cellulose is used for unified explanation in this work.  Question 7: Is it possible to show a chemical modification mechanism via plasma or pertinent chemical equations?Our reply: Thanks for your careful review. The chemical modification mechanism was added and highlighted in Figure 1a (ii). (Page 3/15, Figure 1a) Question 8: Better describe the conditions under which the TGA analysis was performed, that is, flow, type of gas (N2, Air or ???), weighted sample mass.Our reply: Thanks for your comment. The thermogravimetric analysis of the samples were tested from room temperature to 800 ℃ under N2 flow for balance and purge gases. The sample weight of reconstituted tobacco paper-base, PT-reconstituted tobacco paper-base, reconstituted tobacco, and PT-reconstituted tobacco was 2.9242 mg, 4.0215 mg, 6.1214 mg, and 5.2366 mg, respectively. The relevant content was added and highlighted in the revision. (Page 3/15, Characterization section) Question 9: Likewise, better describe how the FT-IR analyzes were carried out.Our reply: Thanks for your careful review. The infrared spectra of reconstituted tobacco paper-base, PT-reconstituted tobacco paper-base, reconstituted tobacco, PT-reconstituted tobacco,  and tobacco extract were both obtained by the Nicolet 8700 Fourier transform infrared (FT-IR) spectrometer. The test wavenumber range of tobacco sheet was from 4000 to 500 cm-1 with membrane measuring reflectance. Moreover, transmittance test of tobacco extract were conducted with a KBr wafer in the wavenumber range varied from 4000 to 400 cm-1. The relevant content was added and highlighted in the revision. (Page 3/15, Characterization section)

Reviewer 3 Report

Please see below my editorial suggestions:

Title:

Line 2                    Plasma Treatment for Cellulose in Tobacco Paper-Base: The Improvement of Surface Hydrophilicity and Mechanical Property

Abstract:

Line 12-15           Should be deleted. If not change “have” to “has”.

                                “The reconstituted tobacco have unparalleled advantages of natural tobacco in reducing harm and coke reduction. Due to the hydrophobicity of the reconstituted tobacco paper-base surface, the coating and absorbing effect of the tobacco extract on the tobacco paper-base is unsatisfactory. Therefore, the improvement on the surface hydrophilicity of the tobacco paper-base has gained increasing attention.”

Introduction:

Line 36                  breeding technology. Thus, it can be used to training Chlorella strains by combining…

Line 63                  Lu et al. showed that..

At the end of this section, there should be a sentence and/or a paragraph where authors indicate the purpose of this study. Please express the objectives of this investigation.

Experimental Section:

Line 85                  Diluted by 50% of what?? Need to indicate the solvent used for dilution.

Need to define the sonication conditions (types of sonication, energy input, T, etc..). Use the same approach as the characterization section.

Size of your specimen??

Results and Discussion:

Line 151                “waveform” or “wavelength”

Figure 5d             There is no captions for the X-axis. Please insert “time”

Figure 9d             There is no captions for the X-axis.

Conclusions:

Well composed.

Reference:

Line 445                                Insert space after the, i.e., 77-81, 88.

                                                Tobacco Sci. Tech. 2019, 52, 8, 77-81, 88.

Author Response

Reply to reviewer #3

Please see below my editorial suggestions:

Question 1: Title:

Line 2: Plasma Treatment for Cellulose in Tobacco Paper-Base: The Improvement of Surface Hydrophilicity and Mechanical Property

Abstract:

Line 12-15: Should be deleted. If not change “have” to “has”.

Our reply: Thanks for your suggestion. The relevant content was deleted. 

Question 2: Introduction:

Line 36: breeding technology. Thus, it can be used to training Chlorella strains by combining…

Line 63: Lu et al. showed that..

At the end of this section, there should be a sentence and/or a paragraph where authors indicate the purpose of this study. Please express the objectives of this investigation.

Our reply: Good comment. “thus it can be used to training Chlorella strains by combining…”. Therefore, the low temperature plasma treatment has been available for agricultural production.  “Lu et al. shows that…”. Therefore, the surface treatment of tobacco paper-base by oxygen plasma to improve hydrophilicity and achieve the acceleration of infiltration rate without decreasing the tobacco extract viscosity and increasing the tobacco paper-base quality is urgently required. The relevant content was added and highlighted in the revision. (Page 1/15 and 2/15, Introduction section)

Question 3: Experimental Section:

Line 85: Diluted by 50% of what?? Need to indicate the solvent used for dilution.

Need to define the sonication conditions (types of sonication, energy input, T, etc..). Use the same approach as the characterization section.

Size of your specimen??

Our reply: Good suggestion. The tobacco extract was diluted by 50% in deionized water. The ultrasonic treatment was performed by automatic ultrasonic cleaning machine (SK5200HP, Shanghai Kedao Ultrasonic Instrument Co., Ltd., China) under 120 W working power and 25 ℃ temperature. The dimension of tobacco paper-base sheet for water/tobacco extract droplet infiltration experiments was 37.3 mm×31.3 mm×0.2 mm. In the quasistatic tensile test, the dimension of tobacco sheet was 20 mm×4 mm×5 mm for the central tensile area (ISO 37: 2005). The relevant content was added and highlighted in the revision. (Page 4/15, Characterization section) 

Question 4: Results and Discussion:

Line 151: “waveform” or “wavelength”

Figure 5d: There is no captions for the X-axis. Please insert “time”

Figure 9d: There is no captions for the X-axis.

Our reply: Thanks for your careful review. The relevant content and images were corrected and highlighted in the revision. (Page 7/15, Figure 5d; Page 11/15, Figure 9d)

Question 5: Conclusions: Well composed.

Our reply: Thank you.

Question 6: Reference:

Line 445: Insert space after the, i.e., 77-81, 88.

        Tobacco Sci. Tech. 2019, 52, 8, 77-81, 88.

Our reply: Thanks for your careful review. The relevant content was revised and highlighted in the revision. (Page 15/15, Reference [26])

[26] Wang, J.; Wen, J.; Zeng, J.; Rao, G.; Yao, P.; Fang, X.; Center, T., Application of flavoring coating technology of reconstituted tobacco base-sheet to cigarette packaging. Tobacco Sci. Tech. 2019, 52, 8, 77-81, 88.
